# Teaching Temporal Logics to Neural Networks[*]

**Christopher Hahn**
CISPA Helmholtz Center for Information Security
Saarbrücken, 66123 Saarland, Germany
`christopher.hahn@cispa.de`

**Frederik Schmitt**
CISPA Helmholtz Center for Information Security
Saarbrücken, 66123 Saarland, Germany
`frederik.schmitt@cispa.de`

**Jens U. Kreber**
Saarland University
Saarbrücken, 66123 Saarland, Germany
`kreber@react.uni-saarland.de`

**Markus N. Rabe**
Google Research
Mountain View, CA, USA
`mrabe@google.com`

**Bernd Finkbeiner**
CISPA Helmholtz Center for Information Security
Saarbrücken, 66123 Saarland, Germany
`finkbeiner@cispa.de`

## Abstract

We study two fundamental questions in neuro-symbolic computing: can deep learning tackle challenging problems in logics end-to-end, and can neural networks learn the semantics of logics. In this work we focus on linear-time temporal logic (LTL), as it is widely used in verification. We train a Transformer on the problem to directly predict a solution, i.e. a trace, to a given LTL formula. The training data is generated with classical solvers, which, however, only provide one of many possible solutions to each formula. We demonstrate that it is sufficient to train on those particular solutions to formulas, and that Transformers can predict solutions even to formulas from benchmarks from the literature on which the classical solver timed out. Transformers also generalize to the semantics of the logics: while they often deviate from the solutions found by the classical solvers, they still predict correct solutions to most formulas.

## 1 Introduction

Machine learning has revolutionized several areas of computer science, such as image recognition (He et al., 2015), face recognition (Taigman et al., 2014), translation (Wu et al., 2016), and board games (Moravcík et al., 2017; Silver et al., 2017). For complex tasks that involve symbolic reasoning, however, deep learning techniques are still considered as insufficient. Applications of deep learning in logical reasoning problems have therefore focused on sub-problems within larger logical frameworks, such as computing heuristics in solvers (Lederman et al., 2020; Balunovic et al., 2018; Selsam & Bjørner, 2019) or predicting individual proof steps (Loos et al., 2017; Gauthier et al., 2018; Bansal et al., 2019; Huang et al., 2018). Recently, however, the assumption that deep learning is not yet ready to tackle hard logical questions was drawn into question. Lample & Charton (2020) demonstrated that Transformer models (Vaswani et al., 2017) perform surprisingly well on symbolic integration, Rabe et al. (2020) demonstrated that self-supervised training leads to mathematical reasoning abilities, and Brown et al. (2020) demonstrated that large-enough language models learn basic arithmetic despite being trained on mostly natural language sources.

This poses the question if other problems that are thought to require symbolic reasoning lend themselves to a direct learning approach. We study the application of Transformer models to challenging

---

[*]Partially supported by the European Research Council (ERC) Grant OSARES (No. 683300) and the Collaborative Research Center "Foundations of Perspicuous Software Systems" (TRR 248, 389792660).

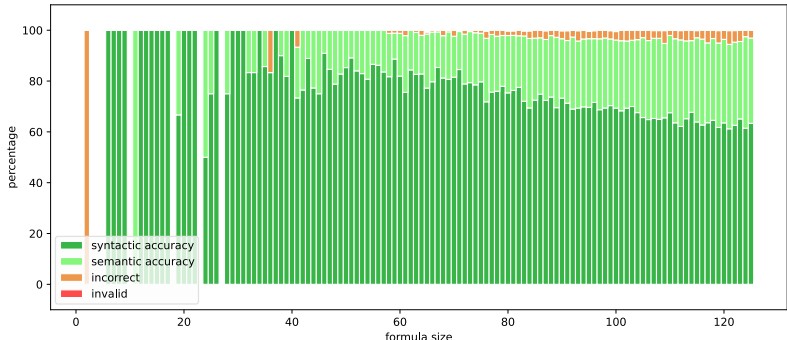

Figure 1: Performance of our best models trained on practical pattern formulas. The x-axis shows the formula size. Syntactic accuracy, i.e., where the Transformer agrees with the generator are displayed in dark green. Instances where the Transformer deviates from the generators output but still provides correct output are displayed in light green; incorrect predictions in orange.

logical problems in verification. We thus consider linear-time temporal logic (LTL) (Pnueli, 1977), which is widely used in the academic verification community (Dwyer et al., 1998; Li et al., 2013; Duret-Lutz et al., 2016; Rozier & Vardi, 2007; Schuppan & Darmawan, 2011; Li et al., 2013; 2014; Schwendimann, 1998) and is the basis for industrial hardware specification languages like the IEEE standard PSL (IEEE-Commission et al., 2005). LTL specifies infinite sequences and is typically used to describe system behaviors. For example, LTL can specify that some proposition $P$ must hold at every point in time ($\square P$) or that $P$ must hold at some future point of time ($\diamondsuit P$). By combining these operators, one can specify that $P$ must occur infinitely often ($\square \diamondsuit P$).

In this work, we apply a direct learning approach to the fundamental problem of LTL to find a satisfying trace to a formula. In applications, solutions to LTL formulas can represent (counter) examples for a specified system behavior, and over the last decades, generations of advanced algorithms have been developed to solve this question automatically. We start from the standard benchmark distribution of LTL formulas, consisting of conjunctions of patterns typically encountered in practice (Dwyer et al., 1998). We then use classical algorithms, notably `spot` by Duret-Lutz et al. (2016), that implement a competitive classical algorithm, to generate solutions to formulas from this distribution and train a Transformer model to predict these solutions directly.

Relatively small Transformers perform very well on this task and we predict correct solutions to 96.8% of the formulas from a held-out test set (see Figure 1). Impressive enough, Transformers hold up pretty well and predict correct solutions in 83% of the cases, even when we focus on formulas on which `spot` timed out. This means that, already today, direct machine learning approaches may be useful to augment classical algorithms in logical reasoning tasks.

We also study two generalization properties of the Transformer architecture, important to logical problems: We present detailed analyses on the generalization to *longer formulas*. It turns out that transformers trained with tree-positional encodings (Shiv & Quirk, 2019) generalize to much longer formulas than they were trained on, while Transformers trained with the standard positional encoding (as expected) do not generalize to longer formulas. The second generalization property studied here is the question whether Transformers learn to imitate the generator of the training data, or whether they learn to solve the formulas according to the semantics of the logics. This is possible, as for most formulas there are many possible satisfying traces. In Figure 1 we highlight the fact that our models often predicted traces that satisfy the formulas, but predict different traces than the one found by the classical algorithm with which we generated the data. Especially when testing the models out-of-distribution we observed that almost no predicted trace equals the solution proposed by the classical solver.

To demonstrate that these generalization behaviors are not specific to the benchmark set of LTL formulas, we also present experimental results on random LTL formulas. Further, we exclude that `spot`, the tool with which we generate example traces, is responsible for these behaviors, by repeating the experiments on propositional formulas for which we generate the solutions by SAT solvers.

The remainder of this paper is structured as follows. We give an overview over related work in Section 2. We describe the problem definitions and present our data generation in Section 3. Our experimental setup is described in Section 4 and our findings in Section 5, before concluding in Section 6.

## 2 RELATED WORK

**Datasets for mathematical reasoning.** While we focus on a classical task from verification, other works have studied datasets derived from automated theorem provers (Blanchette et al., 2016; Loos et al., 2017; Gauthier et al., 2018), interactive theorem provers (Kaliszyk et al., 2017; Bansal et al., 2019; Huang et al., 2018; Yang & Deng, 2019; Polu & Sutskever, 2020; Wu et al., 2020; Li et al., 2020; Lee et al., 2020; Urban & Jakubův, 2020; Rabe et al., 2020), symbolic mathematics (Lample & Charton, 2020), and mathematical problems in natural language (Saxton et al., 2019; Schlag et al., 2019). Probably the closest work to this paper are the applications of Transformers to directly solve differential equations (Lample & Charton, 2020) and directly predict missing assumptions and types of formal mathematical statements (Rabe et al., 2020). We focus on a different problem domain, verification, and demonstrate that Transformers are roughly competitive with classical algorithms in that domain *on their dataset*. Learning has been applied to mathematics long before the rise of deep learning. Earlier works focused on ranking premises or clauses Cairns (2004); Urban (2004; 2007); Urban et al. (2008); Meng & Paulson (2009); Schulz (2013); Kaliszyk & Urban (2014).

**Neural architectures for logical reasoning.** (Paliwal et al., 2020) demonstrate significant improvements in theorem proving through the use of graph neural networks to represent higher-order logic terms. Selsam et al. (2019) presented NeuroSAT, a graph neural network (Scarselli et al., 2008; Li et al., 2017; Gilmer et al., 2017; Wu et al., 2019) for solving the propositional satisfiability problem. In contrast, we apply a generic sequence-to-sequence model to predict the solutions to formulas, not only whether there is a solution. This allows us to apply the approach to a wider set of logics (logics without a CNF). A simplified NeuroSAT architecture was trained for unsat-core predictions (Selsam & Bjørner, 2019). Lederman et al. (2020) have used graph neural networks on CNF to learn better heuristics for a 2QBF solver. Evans et al. (2018) study the problem of logical entailment in propositional logic using tree-RNNs. Entailment is a subproblem of satisfiability and (besides being a classification problem) could be encoded in the same form as our propositional formulas. The formulas considered in their dataset are much smaller than in this work.

**Language models applied to programs.** Transformers have also been applied to programs for tasks such as summarizing code (Fernandes et al., 2018) or variable naming and misuse (Hellendoorn et al., 2020). Other works focused on recurrent neural networks or graph neural networks for code analysis, e.g. (Piech et al., 2015; Gupta et al., 2017; Bhatia et al., 2018; Wang et al., 2018; Allamanis et al., 2017). Another area in the intersection of formal methods and machine learning is the verification of neural networks (Seshia & Sadigh, 2016; Seshia et al., 2018; Singh et al., 2019; Gehr et al., 2018; Huang et al., 2017; Dreossi et al., 2019).

## 3 DATASETS

To demonstrate the generalization properties of the Transformer on logical tasks, we generated several datasets in three different fashions. We will describe the underlying logical problems and our data generation in the following.

### 3.1 TRACE GENERATION FOR LINEAR-TIME TEMPORAL LOGIC

Linear-time temporal logic (LTL, Pnueli, 1977) combines propositional connectives with temporal operators such as the *Next* operator $\bigcirc$ and the *Until* operator $\mathcal{U}$. $\bigcirc \varphi$ means that $\varphi$ holds in the *next* position of a sequence; $\varphi_1 \, \mathcal{U} \, \varphi_2$ means that $\varphi_1$ holds *until* $\varphi_2$ holds. For example, the LTL formula $(b \, \mathcal{U} \, a) \wedge (c \, \mathcal{U} \, \neg a)$ states that $b$ has to hold along the trace until $a$ holds and $c$ has to hold until $a$ does not hold anymore. There also exist derived operators. For example, consider the following specification of an arbiter: $\square(\texttt{request} \rightarrow \lozenge \texttt{grant})$ states that, at every point in time ($\square$-operator), if there is a request signal, then a grant signal must follow at some future point in time ($\lozenge$-operator).

The full semantics and an explanation of the operators can be found in Appendix A. We consider infinite sequences, that are finitely represented in the form of a "lasso" $uv^\omega$, where $u$, called prefix, and $v$, called period, are finite sequences of propositional formulas. We call such sequences *(symbolic) traces*. For example, the symbolic trace $(a \wedge b)^\omega$ defines the infinite sequence where $a$ and $b$ evaluate to true on every position. Symbolic traces allow us to underspecify propositions when they do not matter. For example, the LTL formula $\bigcirc\bigcirc\square a$ is satisfied by the symbolic trace: *true true* $(a)^\omega$, which allow for any combination of propositions on the first two positions.

Our datasets consist of pairs of satisfiable LTL formulas and satisfying symbolic traces generated with tools and automata constructions from the `spot` framework (Duret-Lutz et al., 2016). We use a compact syntax for ultimately periodic symbolic traces: Each position in the trace is separated by the delimiter ";". True and False are represented by "1" and "0", respectively. The beginning of the period $v$ is signaled by the character "{" and analogously its end by "}". For example, the ultimately periodic symbolic trace denoted by $a; a; a; \{b\}$, describes all infinite traces where on the first 3 positions $a$ must hold followed by an infinite period on which $b$ must hold on every position.

Given a satisfiable LTL formula $\varphi$, our trace generator constructs a Büchi automaton $A_\varphi$ that accepts exactly the language defined by the LTL formula, i.e., $\mathcal{L}(A_\varphi) = \mathcal{L}(\varphi)$. From this automaton, we construct an arbitrary accepted symbolic trace, by searching for an accepting run in $A_\varphi$.

### 3.1.1 SPECIFICATION PATTERN

Our main dataset is constructed from formulas following 55 LTL specification patterns identified by the literature (Dwyer et al., 1998). For example, the arbiter property $(\Diamond p_0) \to (p_1 \, \mathcal{U} \, p_0)$, stating that if $p_0$ is scheduled at some point in time, $p_1$ is scheduled until this point. The largest specification pattern is of size 40 consisting of 6 atomic propositions. It has been shown that conjunctions of such patterns are challenging for LTL satisfiability tools that rely on classical methods, such as automata constructions (Li et al., 2013). They start coming to their limits when more than 8 pattern formulas are conjoined. We decided to build our dataset in a similar way from these patterns only to allow for a better comparison.

We conjoined random specification patterns with randomly chosen variables (from a supply of 6 variables) until one of the following four conditions are met: 1) the formula size succeeds 126, 2) more than 8 formulas would be conjoined, 3) our automaton-based generator timed out ($> 1s$) while computing the solution trace, or 4) the formula would become unsatisfiable. In total, we generated 1664487 formula-trace pairs in 24 hours on 20 CPUs. While generating, approximately 41% of the instances ran into the first termination condition, 21% into the second, 37% into the third and 1% into the fourth. We split this set into an 80% training set, a 10% validation set, and a 10% test set. The size distribution of the dataset can be found in Appendix B.

For studying how the Transformer performs on longer specification patterns, we accumulated pattern formulas where `spot` timed out ($> 60s$) while searching for a satisfying trace. We call this dataset $LTLUnsolved254$. We capped the maximum length at 254, which is twice as large as the formulas the model saw during training. The size distribution of the generated formulas can be found in Appendix B.

In the following table, we illustrate the complexity of our training dataset with two examples from the above described set $LTLPattern126$, where the subsequent number of the notation of our datasets denotes the maximum size of a formula's syntax tree. The first line shows the LTL formula and the symbolic trace in mathematical notation. The second line shows the input and output representation of the Transformer (in Polish notation):

| LTL formula | satisfying symbolic trace |
|---|---|
| $\square(a \to \Diamond d) \wedge \neg f \, \mathcal{W} \, f \, \mathcal{W} \, \neg f \, \mathcal{W} \, f \, \mathcal{W} \, \square \neg f$ $\wedge (\Diamond c \to \neg c \mathcal{U}(c \wedge \neg b \, \mathcal{W} \, b \, \mathcal{W} \, \neg b \, \mathcal{W} \, b \, \mathcal{W} \, \square \neg b))$ | $(\neg a \wedge \neg c \wedge \neg f \vee \neg c \wedge d \wedge \neg f)^\omega$ |
| `&&G>aFdW!fWfW!fWfG!f>FcU!c&cW!bWbW!bWbG!b` | `{!a&!c&!f|!c&d&!f}` |
| $\square(b \wedge \neg a \wedge \Diamond a \to c \mathcal{U} a) \wedge \square(a \to \square \neg c) \wedge (\Diamond b \to \neg b$ $\mathcal{U}(b \wedge \neg f \, \mathcal{W} \, f \, \mathcal{W} \, \neg f \, \mathcal{W} \, f \, \mathcal{W} \, \square \neg f)) \wedge (\Diamond a \to (c \wedge \bigcirc(\neg a \mathcal{U} e)$ $\to \bigcirc(\neg a \mathcal{U}(e \wedge \Diamond f))) \mathcal{U} a) \wedge \Diamond c \wedge \square(a \square \Diamond e \to \neg(\neg e \wedge f \wedge \bigcirc$ $(\neg e \mathcal{U}(\neg e \wedge d))) \mathcal{U}(e \vee c)) \wedge (\square \neg a \vee \Diamond(a \wedge \neg f \, \mathcal{W} \, d)) \wedge \square(e \to \square \neg c)$ | $(\neg a \wedge b \wedge \neg c \wedge \neg e \wedge f)(\neg a \wedge \neg c$ $\wedge \neg e \wedge \neg f)(\neg a \wedge \neg c \wedge \neg e \wedge f)$ $(\neg a \wedge c \wedge \neg e \wedge \neg f)(\neg a \wedge \neg e \wedge \neg f)^\omega$ |
| `&&&&&&&G>&&b!aFaUcaG>aGc>FbU!b&bW!fWfW!fW` `fG!f>FaU>&cXU!aeXU!a&eFfaFcG>&aFeU!&` `&!efXU!e&!ed|ec!G!aF&aW!fdG>eG!c` | `&&&&!ab!c!ef;&&&!a!c!e!f;` `&&&!a!c!ef;&&&!ac!e!f` `;{&&!a!e!f}` |

### 3.1.2 RANDOM FORMULAS

To show that the generalization properties of the Transformer are not specific to our data generation, we also generated a dataset of random formulas. Our dataset of random formulas consist of 1 million generated formulas and their solutions, i.e., a satisfying symbolic trace. The number of different propositions is fixed to 5. Each dataset is split into a training set of 800K formulas, a validation set of 100K formulas, and a test set of 100K formulas. All datasets are uniformly distributed in size, apart from the lower-sized end due to the limited number of unique small formulas. The formula and trace distribution of the dataset $LTLRandom35$, as well as three randomly drawn example instances can be found in Appendix B. Note that we filtered out examples with traces larger than 62 (less than 0.05% of the original set).

To generate the formulas, we used the `randltl` tool of the `spot` framework, which builds unique formulas in a specified size interval, following a supplied node probability distribution. During the building process, the actual distribution occasionally differs from the given distribution in order to meet the size constraints, e.g., by masking out all binary operators. The distribution between all $k$-ary nodes always remains the same. To furthermore achieve a (quasi) uniform distribution in size, we subsequently filtered the generated formulas. Our node distribution puts equal weight on all operators $\neg, \wedge, \bigcirc$ and $\mathcal{U}$. Constants `True` and `False` are allowed with 2.5 times less probability than propositions.

### 3.2 ASSIGNMENT GENERATION FOR PROPOSITIONAL LOGIC

To show that the generalization of the Transformer to the semantics of logics is not a unique attribute of LTL, we also generated a dataset for propositional logic (SAT). A propositional formula consists of Boolean operators $\wedge$ (and), $\vee$ (or), $\neg$ (not), and variables also called literals or propositions. We consider the derived operators $\varphi_1 \rightarrow \varphi_2 \equiv \neg\varphi_1 \vee \varphi_2$ (implication), $\varphi_1 \leftrightarrow \varphi_2 \equiv (\varphi_1 \rightarrow \varphi_2) \wedge (\varphi_2 \rightarrow \varphi_1)$ (equivalence), and $\varphi_1 \oplus \varphi_2 \equiv \neg(\varphi_1 \leftrightarrow \varphi_2)$ (xor). Given a propositional Boolean formula $\varphi$, the satisfiability problem asks if there exists a Boolean assignment $\Pi : \mathcal{V} \mapsto \mathbb{B}$ for every literal in $\varphi$ such that $\varphi$ evaluates to $true$. For example, consider the following propositional formula, given in conjunctive normal form (CNF): $(x_1 \vee x_2 \vee \neg x_3) \wedge (\neg x_1 \vee x_3)$. A possible satisfying assignment for this formula would be $\{(x_1, true), (x_2, false), (x_3, true)\}$. We allow a satisfying assignment to be *partial*, i.e., if the truth value of a propositions can be arbitrary, it will be omitted. For example, $\{(x_1, true), (x_3, true)\}$ would be a satisfying partial assignment for the formula above. We define a *minimal unsatisfiable core* of an unsatisfiable formula $\varphi$, given in CNF, as an unsatisfiable subset of clauses $\varphi_{core}$ of $\varphi$, such that every proper subset of clauses of $\varphi_{core}$ is still satisfiable.

We, again, generated 1 million random formulas. For the generation of propositional formulas, the specified node distribution puts equal weight on $\wedge$, $\vee$, and $\neg$ operators and half as much weight on the derived operators $\leftrightarrow$ and $\oplus$ individually. In contrast to previous work (Selsam et al., 2019), which is restricted to formulas in CNF, we allow an arbitrary formula structure and derived operators.

A satisfying assignment is represented as an alternating sequence of propositions and truth values, given as 0 and 1. The sequence $a0b1c0$, for example, represents the partial assignment $\{(a, false), (b, true), (c, false)\}$, meaning that the truth values of propositions $d$ and $e$ can be chosen arbitrarily (note that we allow five propositions). We used pyaiger (Vazquez-Chanlatte, 2018), which builds on Glucose 4 (Audemard & Simon, 2018) as its underlying SAT solver. We construct the partial assignments with a standard method in SAT solving: We query the SAT solver for a minimal unsatisfiable core of the *negation* of the formula. To give the interested reader an idea of the level of difficulty of the dataset, the following table shows three random examples from our training set $PropRandom35$. The first line shows the formula and the assignment in mathematical notation. The second line shows the syntactic representation (in Polish notation):

| propositional formula | satisfying partial assignment |
|---|---|
| $((d \wedge \neg e) \wedge (\neg a \vee \neg e)) \leftrightarrow ((\neg \oplus (\neg b \leftrightarrow \neg e))$ $\vee((e \oplus (b \wedge d)) \oplus \neg(\neg c \vee (\neg a \leftrightarrow e))))$ | $\{(a, 0), (b, 0), (c, 1), (d, 1), (e, 0)\}$ |
| `<->&&d!e|!a!e|xor!b<->!b!exorxore&bd!|!c<->!ae` | `a0b0c1d1e0` |
| $(c \vee e) \vee (\neg a \leftrightarrow \neg b)$ | $\{(c, 1)\}$ |
| `||ce<->!a!b` | `c1` |
| $\neg((b \vee e) \oplus ((\neg a \vee (\neg d \leftrightarrow \neg e))$ $\vee(\neg b \vee (((\neg a \wedge b) \wedge \neg b) \wedge d))))$ | $\{(d, 1), (e, 1)\}$ |
| `!xor!be||!a<->!d!e||!b&&&!ab!b!d` | `d1e1` |

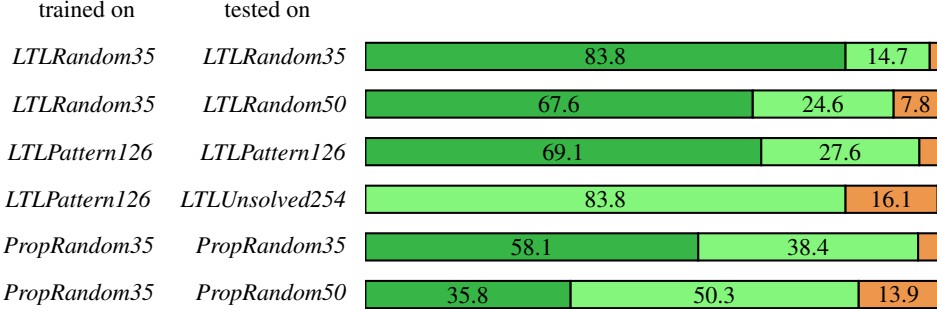

Figure 2: Overview of our main experimental results: the performance of our best performing models on our different datasets. The percentage of a dark green bar refers to the syntactic accuracy, the percentage of a light green bar to the semantic accuracy without the syntactic accuracy, and the incorrect predictions are visualized in orange.

To test the Transformer on even more challenging formulas, we constructed a dataset of CNF formulas using the generation script of Selsam et al. (2019) from their publicly available implementation. A random CNF formula is built by adding clauses until the addition of a further clause would lead to an unsatisfiable formula. We used the parameters $p_{geo} = 0.9$ and $p_{k2} = 0.75$ to generate formulas that contain up to 15 variables and have a maximum size of 250. We call this dataset $PropCNF250$.

## 4 EXPERIMENTAL SETUP

We have implemented the Transformer architecture (Vaswani et al., 2017). Our implementation processes the input and output sequences token-by-token. We trained on a single GPU (NVIDIA P100 or V100). All training has been done with a dropout rate of 0.1 and early stopping on the validation set. Note that the embedding size will automatically be floored to be divisible by the number of attention heads. The training of the best models took up to 50 hours. For the output decoding, we utilized a beam search (Wu et al., 2016), with a beam size of 3 and an $\alpha$ of 1.

Since the solution of a logical formula is not necessarily unique, we use two different measures of accuracy to evaluate the generalization to the semantics of the logics: we distinguish between the *syntactic* accuracy, i.e., the percentage where the Transformers prediction syntactically matches the output of our generator and the *semantic* accuracy, i.e., the percentage where the Transformer produced a different solution. We also differentiate between incorrect predictions and syntactically invalid outputs which, in fact, happens only in 0.1% of the cases in $LTLUnsolved254$.

In general, our best performing models used 8 layers, 8 attention heads, and an FC size of 1024. We used a batch size of 400 and trained for $450K$ steps (130 epochs) for our specification pattern dataset, and a batch size of 768 and trained for $50K$ steps (48 epochs) for our random formula dataset. A hyperparameter study can be found in Appendix C.

## 5 EXPERIMENTAL RESULTS

In this section, we describe our experimental results. First, we show that a Transformer can indeed solve the task of providing a solution, i.e., a trace for a linear-time temporal logical (LTL) formula. For this, we describe the results from training on the dataset $LTLPattern126$ of specification patterns that are commonly used in the context of verification. Secondly, we show two generalization properties that the Transformer evinces on logic reasoning tasks: 1) the generalization to larger formulas (even so large that our data generator timed out) and 2) the generalization to the semantics of the logic. We strengthen this observation by considering a different dataset of random LTL formulas. Thirdly, we provide results for a model trained on a different logic and with a different data generator. We thereby demonstrate that the generalization behaviors of the Transformer are not specific to LTL and the LTL solver implemented with spot that we used to generate the data. An overview of our training results is displayed in Figure 2.

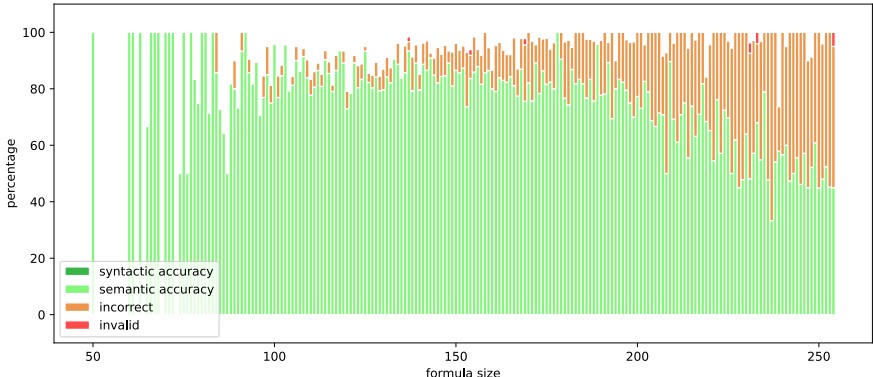

Figure 3: Predictions of our best performing model, trained on $LTLUnsolved254$, on 5704 specification patterns for which spot timed out ($> 60s$). Semantic accuracy is displayed in green; incorrect traces in orange; syntactically invalid traces in red.

## 5.1 SOLVING LINEAR-TIME TEMPORAL LOGICAL FORMULAS

We trained a Transformer on our specification on $LTLPattern126$. Figure 1 in the introduction displays the performance of our best model on this dataset. We observed a syntactic accuracy of 69.1% and a semantic accuracy of 96.8%. With this experiment we can already deduce that it seems easier for the Transformer to learn the underlying semantics of LTL than to learn the particularities of the generator. Further we can see that as the formula length grows, the syntactic accuracy begins to drop. However, that drop is much smaller in the semantic accuracy—the model still mostly predicts correct traces for long formulas.

As a challenging benchmark, we tested our best performing model on $LTLUnsolved254$. It predicted correct solutions in 83% of the cases, taking on average 15$s$ on a single CPU. The syntactic accuracy is 0% as there was no output produced by spot within the timeout. The results of the experiments are visualized in Figure 3. Note that this does not mean that our Transformer models necessariy outperform classical algorithms across the board. However, since *verifying* solutions to LTL formulas is much easier than *finding* solutions (AC[1](logDCFL) vs PSPACE), this experiment shows that the predictions of a deep neural network can be a valuable extension to the verification tool box.

## 5.2 GENERALIZATION PROPERTIES

To prove that the generalization to the semantics is independent of the data generation, we also trained a model on a dataset of randomly generated formulas. The *unshaded* part of Figure 4 displays the performance of our best model on the $LTLRandom35$ dataset. The Transformers were solely trained on formulas of size less or equal to 35. We observe that in this range the exact syntactic accuracy decreases when the formulas grow in size. The semantic accuracy, however, stays, again, high. The model achieves a syntactic accuracy of 83.8% and a semantic accuracy of 98.5% on $LTLRandom35$, i.e., in 14.7% of the cases, the Transformer deviates from our automaton-based data generator. The evolution of the syntactic and the semantic accuracy during training can be found in Appendix D.

To show that the generalization to larger formulas is independent from the data generation method, we also tested how well the Transformer generalizes to randomly generated LTL formulas of a size it has never seen before. We used our model trained on $LTLRandom35$ and observed the performance on $LTLRandom50$. The model preserves the semantic generalization, displayed in the *shaded* part of Figure 4. It outputs exact syntactic matches in 67.6% of the cases and achieves a semantic accuracy of 92.2%. For the generalization to larger formulas we utilized a positional encoding based on the tree representation of the formula (Shiv & Quirk, 2019). When using the

standard positional encoding instead, the accuracy drops, as expected, significantly. A visualization of this experiments can be found in Appendix E.

In a further experiment, we tested the out-of-distribution (OOD) generalization of the Transformer on the trace generation task. We generated a new dataset $LTLRandom126$ to match the formula sizes and the vocabulary of $LTLPattern126$. A model trained on $LTLRandom126$ achieves a semantic accuracy of $24.7\%$ (and a syntactic accuracy of only $1.0\%$) when tested on $LTLPattern126$. Vice versa, a model trained on $LTLPattern126$ achieves a semantic accuracy of $38.5\%$ (and a semantic accuracy of only $0.5\%$) when tested on $LTLRandom126$. Testing the models OOD increases the gap between syntactic and semantic correctness dramatically. This underlines that the models learned the nature of the LTL semantics rather than the generator process. Note that the two distributions are very different.

Following these observations, we also tested the performance of our models on other patterns from the literature. We observe a higher semantic accuracy for our model trained on random formulas and a higher gap between semantic and syntactic accuracy for our model trained on pattern formulas:

| Patterns | Number of Patterns | Trained on | Syn. Acc. | Sem. Acc. |
|---|---|---|---|---|
| dac (Dwyer et al., 1998) | 55 | $LTLRandom126$ | 49.1% | 81.8% |
| eh (Etessami & Holzmann, 2000) | 11 | $LTLRandom126$ | 81.8% | 90.9% |
| hkrss (Holeček et al., 2004) | 49 | $LTLRandom126$ | 71.4% | 83.7% |
| p (Pelánek, 2007) | 20 | $LTLRandom126$ | 65.0% | 90.0% |
| eh (Etessami & Holzmann, 2000) | 11 | $LTLPattern126$ | 0.0% | 36.4% |
| hkrss (Holeček et al., 2004) | 49 | $LTLPattern126$ | 14.3% | 49.0% |
| p (Pelánek, 2007) | 20 | $LTLPattern126$ | 10.0% | 60.0% |

In a last experiment on LTL, we tested the performance of our models on handcrafted formulas. We observed that formulas with multiple until statements that describe overlapping intervals were the most challenging. This is no surprise as these formulas are the source of PSPACE-hardness of LTL.

| $a\,\mathcal{U}\,b \wedge a\,\mathcal{U}\,\neg b$ | $(a \wedge \neg b)\,(b)\,(true)^{\omega}$ |
|---|---|
| `&UabUa!b` | `&a!b;b;{1}` |

While the above formula can be solved by most models, when scaling this formula to four overlapping until intervals, all of our models fail: For example, a model trained on $LTLRandom35$ predicted the trace $(a \wedge b \wedge c)\,(a \wedge \neg b \wedge \neg c)\,(b \wedge c)\,(true)^{\omega}$, which does not satisfy the LTL formula.

| $(a\,\mathcal{U}\,b \wedge c) \wedge (a\,\mathcal{U}\,\neg b \wedge c) \wedge (a\,\mathcal{U}\,b \wedge \neg c) \wedge (a\,\mathcal{U}\,\neg b \wedge \neg c)$ | $(a \wedge b \wedge c)\,(a \wedge \neg b \wedge \neg c)\,(b \wedge c)\,(true)^{\omega}$ |
|---|---|
| `&&&Ua&bcUa&!bcUa&b!cUa&!b!c` | `&&abc;&&a!b!c;&bc;1` |

## 5.3 PREDICTING ASSIGNMENTS FOR PROPOSITIONAL LOGIC

To show that the generalization to the semantic is not a specific property of LTL, we trained a Transformer to solve the assignment generation problem for propositional logic, which is a substantially different logical problem.

As a baseline for our generalization experiments on propositional logic, we trained and tested a Transformer model with the following hyperparameter on $PropRandom35$:

| Embedding size | Layers | Heads | FC size | Batch Size | Train Steps | Syn. Acc. | Sem. Acc. |
|---|---|---|---|---|---|---|---|
| enc:128, dec:64 | 6 | 6 | 512 | 1024 | 50K | 58.1% | **96.5%** |

We observe a striking $38.4\%$ gap between predictions that were syntactical matches of our DPLL-based generator and correct predictions of the Transformer. Only $3.5\%$ of the time, the Transformer outputs an incorrect assignment. Note that we allow the derived operators $\oplus$ and $\leftrightarrow$ in these experiments, which succinctly represent complicated logical constructs.

The formula $b \vee \neg(a \wedge d)$ occurs in our dataset $PropRandom35$ and its corresponding assignment is $\{(a, 0)\}$. The Transformer, however, outputs $\texttt{d0}$, i.e., it goes with the assignment of setting $d$ to *false*, which is also a correct solution. A visualization of this example can be found in Appendix F. When the formulas get larger, the solutions where the Transformer differs from the DPLL algorithm accumulate. Consider, for example, the formula $\neg b \vee (e \leftrightarrow b \vee c \vee \neg d) \vee (c \wedge (b \oplus (a \oplus \neg d)) \oplus (\neg c \leftrightarrow d) \wedge (a \leftrightarrow (b \oplus (b \oplus e))))$, which is also in the dataset $PropRandom35$. The generator suggests

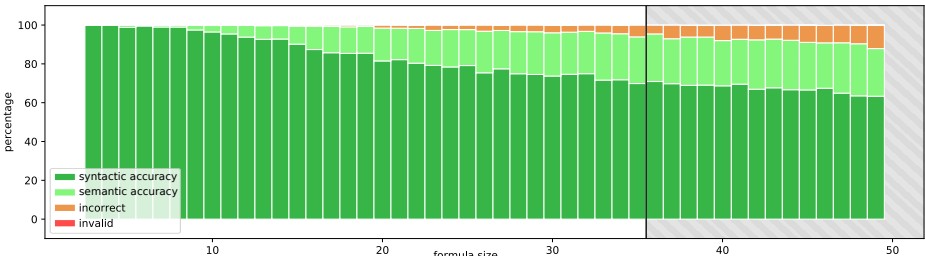

Figure 4: Syntactic and semantic accuracy of our best performing model (only trained on *LTLRandom35*) on *LTLRandom50*. Dark green is syntactically correct; light green is semantically correct, orange is incorrect.

the assignment $\{(a, 1), (c, 1), (d, 0)\}$. The Transformer, however, outputs $e0$, i.e., the singleton assignment of setting $e$ to *false*, which turns out to be a (very small) solution as well.

We achieved stable training in this experiment by setting the decoder embedding size to either 64 or even 32. Keeping the decoder embedding size at 128 led to very unstable training.

We also tested whether the generalization to the semantics is preserved when the Transformer encounters propositional formulas of a larger size than it ever saw during training. We, again, utilized the tree positional encoding. When challenged with formulas of size 35 to 50, our best performing model trained on $PropRandom35$ achieves a syntactic accuracy of $35.8\%$ and a semantic accuracy of $86.1\%$. In comparison, without the tree positional encoding, the Transformer achieves a syntactic match of only $29.0\%$ and an overall accuracy of only $75.7\%$. Note that both positional encodings work equally well when not considering larger formulas.

In a last experiment, we tested how the Transformer performs on more challenging propositional formulas in CNF. We thus trained a model on $PropCNF250$, where it achieved a semantic accuracy of $65.1\%$ and a syntactic accuracy of $56.6\%$. We observe a slightly lower gap compared to our LTL experiments. The Transformer, however, still deviates even on such formulas from the generator.

## 6 Conclusion

We trained a Transformer to predict solutions to linear-time temporal logical (LTL) formulas. We observed that our trained models evince powerful generalization properties, namely, the generalization to the semantics of the logic, and the generalization to larger formulas than seen during training. We showed that these generalizations do *not* depend on the underlying logical problem nor on the data generator. Regarding the performance of the trained models, we observed that they can compete with classical algorithms for generating solutions to LTL formulas. We built a test set that contained only formulas that were generated out of practical verification patterns, on which even our data generator timed out. Our best performing model, although it was trained on much smaller formulas, predicts correct traces $83\%$ of the time.

The results of this paper suggest that deep learning can already augment combinatorial approaches in automatic verification and the broader formal methods community. With the results of this paper, we can, for example, derive novel algorithms for trace generation or satisfiability checking of LTL that first query a Transformer for trace predictions. These predictions can be checked efficiently. Classical methods can serve as a fall back or check partial solutions providing guidance to the Transformer. The potential that arises from the advent of deep learning in logical reasoning is immense. Deep learning holds the promise to empower researchers in the automated reasoning and formal methods communities to make bigger jumps in the development of new automated verification methods, but also brings new challenges, such as the acquisition of large amounts of data.

### Acknowledgements

We thank Christian Szegedy, Jesko Hecking-Harbusch, and Niklas Metzger for their valuable feedback on an earlier version of this paper.

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

APPENDIX

## A   LINEAR-TIME TEMPORAL LOGIC (LTL)

In this section, we provide the formal syntax and semantics of Linear-time Temporal Logic (LTL). The formal syntax of LTL is given by the following grammar:

$$\varphi ::= p \mid \neg\varphi \mid \varphi \wedge \varphi \mid \bigcirc\varphi \mid \varphi\,\mathcal{U}\,\varphi,$$

where $p \in AP$ is an atomic proposition. Let $AP$ be a set of *atomic propositions*. A (*explicit*) *trace* $t$ is an infinite sequence over subsets of the atomic propositions. We define the set of traces $TR := (2^{AP})^\omega$. We use the following notation to manipulate traces: Let $t \in TR$ be a trace and $i \in \mathbb{N}$ be a natural number. With $t[i]$ we denote the set of propositions at $i$-th position of $t$. Therefore, $t[0]$ represents the starting element of the trace. Let $j \in \mathbb{N}$ and $j \geq i$. Then $t[i, j]$ denotes the sequence $t[i]\, t[i+1] \dots t[j-1]\, t[j]$ and $t[i, \infty]$ denotes the infinite suffix of $t$ starting at position $i$.

Let $p \in AP$ and $t \in TR$. The semantics of an LTL formula is defined as the smallest relation $\models$ that satisfies the following conditions:

$$
\begin{array}{lll}
t \models p & \text{iff} & p \in t[0] \\
t \models \neg\varphi & \text{iff} & t \not\models \varphi \\
t \models \varphi_1 \wedge \varphi_2 & \text{iff} & t \models \varphi_1 \text{ and } t \models \varphi_2 \\
t \models \bigcirc\varphi & \text{iff} & t[1, \infty] \models \varphi \\
t \models \varphi_1\,\mathcal{U}\,\varphi_2 & \text{iff} & \text{there exists } i \geq 0 : t[i, \infty] \models \varphi_2 \\
& & \text{and for all } 0 \leq j < i \text{ we have } t[j, \infty] \models \varphi_1
\end{array}
$$

There are several derived operators, such as $\Diamond\varphi \equiv true\,\mathcal{U}\,\varphi$ and $\Box\varphi \equiv \neg\Diamond\neg\varphi$. $\Diamond\varphi$ states that $\varphi$ will *eventually* hold in the future and $\Box\varphi$ states that $\varphi$ holds *globally*. Operators can be nested: $\Box\Diamond\varphi$, for example, states that $\varphi$ has to occur infinitely often.

## B   SIZE DISTRIBUTION IN THE DATASETS

In this section, we provide insight into the size distribution of our datasets. Figure 5 shows the size distribution of the formulas in our dataset $LTLPattern126$.

Figure 6 shows the size distribution of our generated formulas and their traces in the dataset $LTLRandom35$. Table 1 shows three randomly drawn example instances of the dataset $LTLRandom35$.

Lastly, Figure 7 shows the size distribution of formulas in our dataset $LTLUnsolved254$.

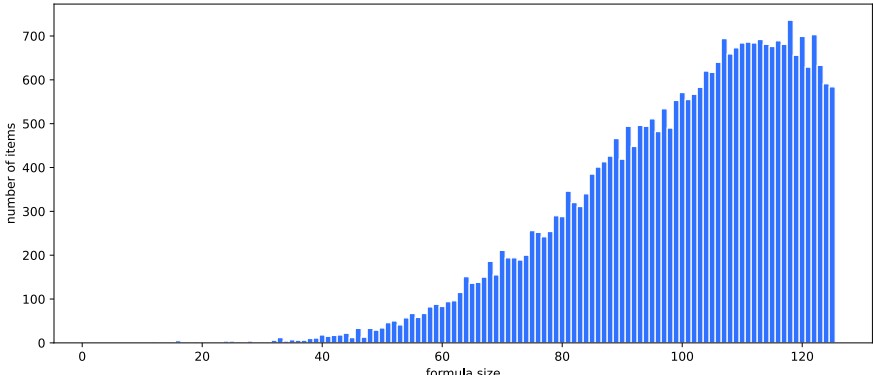

Figure 5: Size distributions in the $LTLPattern126$ test set: on the x-axis is the size of the formulas; on the y-axis the number of formulas.

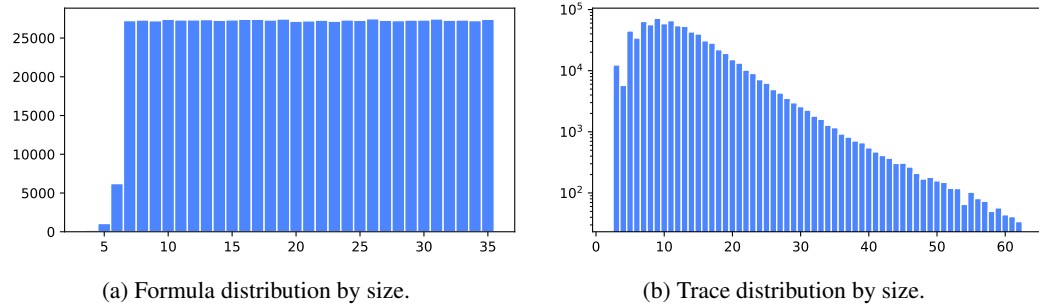

(a) Formula distribution by size.          (b) Trace distribution by size.

Figure 6: Size distributions in the $LTLRandom35$ training set: on the x-axis is the size of the formulas/traces; on the y-axis the number of formulas/traces.

Table 1: Three random examples from $LTLRandom35$ training set. The first line shows the LTL formula and the symbolic trace in mathematical notation. The second line shows the syntactic representation (in Polish notation):

| LTL formula | satisfying symbolic trace |
|---|---|
| $\bigcirc((d\,\mathcal{U}\,c)\,\mathcal{U}\bigcirc\bigcirc d)\wedge\bigcirc(b\wedge\neg(\neg d\,\mathcal{U}\,c))$ | $true\ (b\wedge\neg c\wedge\neg d)\ (\neg c\wedge d)\ d\ (true)^{\omega}$ |
| `&XUUdcXXdX&b!U!dc` | `1;&&b!c!d;&!cd;d;{1}` |
| $\neg\bigcirc((\bigcirc e\wedge(true\,\mathcal{U}\,b)\wedge\bigcirc c)\,\mathcal{U}\,c)$ | $true\ (\neg b\wedge\neg c)\ (\neg b)^{\omega}$ |
| `!XU&&XeU1bXcc` | `1;&!b!c;{!b}` |
| $\bigcirc\neg((\neg c\wedge d)\,\mathcal{U}\bigcirc d)$ | $true\ (c\vee\neg d)\ (\neg d)\ (true)^{\omega}$ |
| `X!U&!cdXd` | `1;|c!d;!d;{1}` |

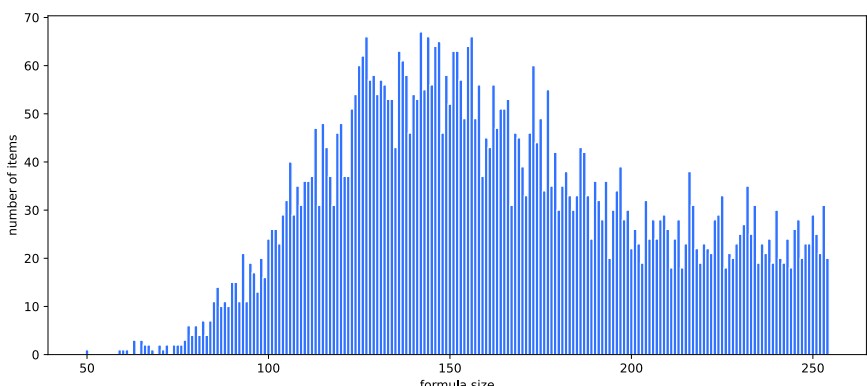

Figure 7: Size distributions in the $LTLUnsolved254$ test set: on the x-axis is the size of the formulas; on the y-axis the number of formulas.

## C    HYPERPARAMETER ANALYSIS

Table 2 shows the effect of the most significant parameters on the performance of Transformers. The performance largely benefits from an increased number of layers, with $8$ yielding the best results. Increasing the number further, even with much more training time, did not result in better or even led to worse results. A slightly less important role plays the number of heads and the dimension of the intermediate fully-connected feed-forward networks (FC). While a certain FC size is important, increasing it alone will not improve results. Changing the number of heads alone has also almost no impact on performance. Increasing both simultaneously, however, will result in a small gain.

Table 2: Syntactic accuracy and semantic accuracy of different Transformers, tested on *LTLRandom*35: Layers refer to the size of the encoder and decoder stacks; Heads refer to the number of attention heads; FC size refers to the size of the fully-connected neural networks inside the encoder and decoders.

| Embedding size | Layers | Heads | FC size | Batch Size | Train Steps | Syn. Acc. | Sem. Acc. |
|---|---|---|---|---|---|---|---|
| 128 | 3 | 4 | 512 | 512 | 45K | 78.0% | 97.1% |
| 128 | 5 | 2 | 512 | 512 | 45K | 80.4% | 97.4% |
| 128 | 5 | 4 | 256 | 512 | 45K | 81.0% | 97.4% |
| 128 | 5 | 4 | 512 | 512 | 45K | 82.0% | 97.9% |
| 128 | 5 | 4 | 1024 | 512 | 45K | 80.3% | 97.3% |
| 128 | 5 | 6 | 1024 | 512 | 45K | 81.8% | 97.7% |
| 128 | 5 | 8 | 512 | 512 | 45K | 82.0% | 97.8% |
| 128 | 5 | 8 | 1024 | 512 | 45K | 82.5% | 97.9% |
| 128 | 5 | 8 | 1500 | 512 | 45K | 82.6% | 97.8% |
| 128 | 5 | 12 | 1024 | 512 | 45K | 81.9% | 97.5% |
| 128 | 8 | 4 | 512 | 512 | 45K | 83.2% | 98.3% |
| 128 | 8 | 8 | 1024 | 768 | 50K | 83.8% | **98.5%** |
| 128 | 10 | 4 | 512 | 512 | 75K | 82.9% | 97.6% |
| 256 | 5 | 4 | 512 | 512 | 45K | 82.3% | 97.9% |

This seems reasonable, since more heads can provide more distinct information to the subsequent processing by the fully-connected feed-forward network. Increasing the embeddings size from 128 to 256 very slightly improves the syntactic accuracy. But likewise it also degrades the semantic accuracy, so we therefore stuck with the former setting.

## D   ACCURACY DURING TRAINING

In Figure 8 we show the evolution of both the syntactic accuracy and the semantic accuracy during the training process. Note the significant difference right from the beginning. This demonstrates the importance of a suitable performance measure when evaluating machine learning algorithms on logical reasoning tasks.

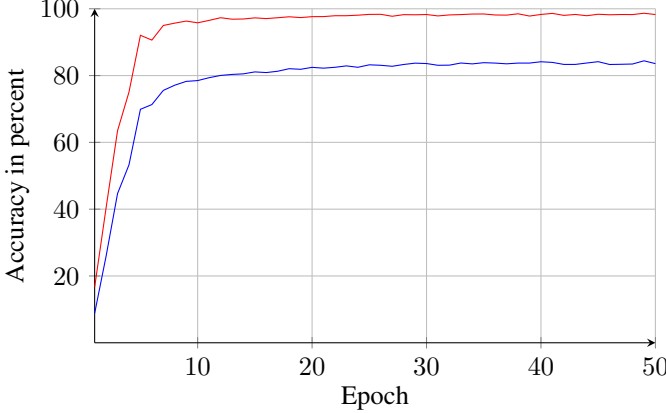

Figure 8: Syntactic accuracy (blue) and semantic accuracy (red) of our best performing model, evaluated on a subset of 5K samples of *LTLRandom*35 per epoch.

# E  DIFFERENT POSITIONAL ENCODINGS

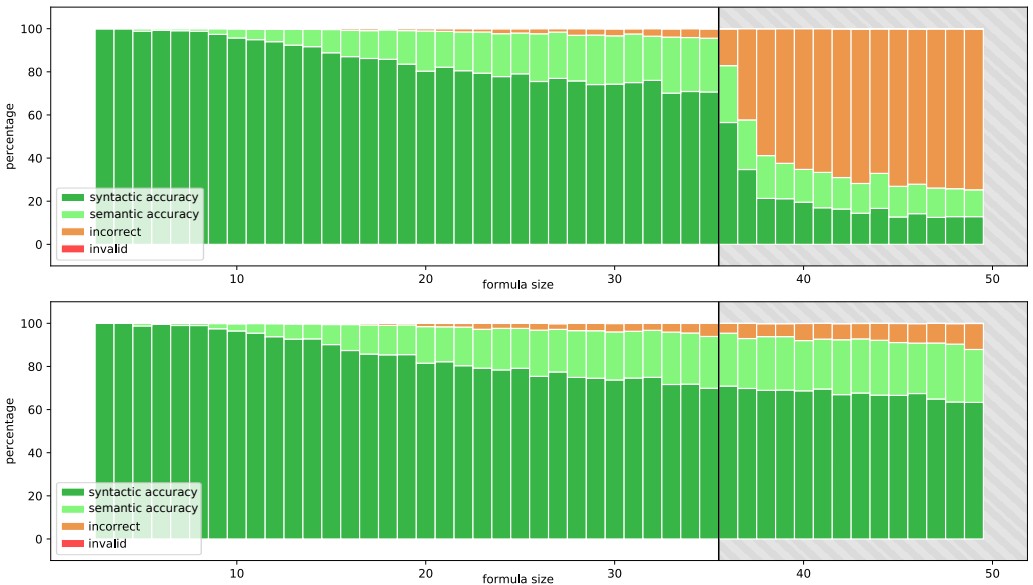

Figure 9: Performance of our best model (only trained on $LTLRandom35$) on $LTLRandom50$ with a standard positional encoding (top) and a tree positional encoding (bottom). The syntactic accuracy is displayed in green, the semantic accuracy in light green and the incorrect predictions in orange. The shaded area indicates the formula sizes the model was not trained on.

# F  HANDCRAFTED EXAMPLES

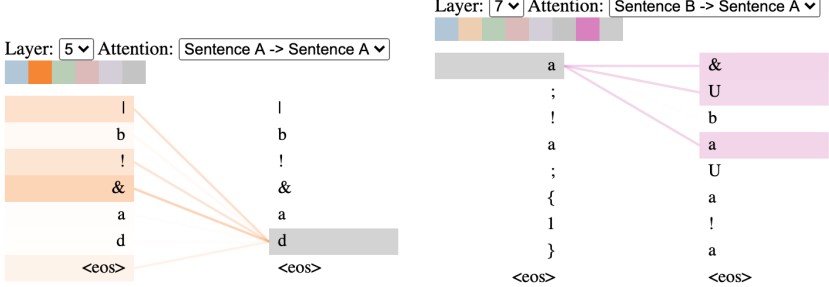

Figure 10: Self-attention of the example propositional formula $b \vee \neg(a \wedge d)$ in dataset $PropRandom35$ (left). Encoder-decoder-attention of the example LTL formula $(b\,\mathcal{U}\,a) \wedge (a\,\mathcal{U}\,\neg a)$ in dataset $LTLRandom35$ (right).

The LTL formula $(b\,\mathcal{U}\,a) \wedge (a\,\mathcal{U}\,\neg a)$ states that $b$ has to hold along the trace until $a$ holds and $a$ has to hold until $a$ does not hold anymore. The automaton-based generator suggests the trace $(\neg a \wedge b)\,a\,(true)^\omega$, i.e., to first satisfy the second until by immediately disallowing $a$. The satisfaction of the first until is then postponed to the second position of trace, which forces $b$ to hold on the first position. The Transformer, however, chooses the following more general trace $a\,(\neg a)\,(true)^\omega$, by satisfying the until operators in order (see Figure 10).

