# OpenReview forum: "Teaching Temporal Logics to Neural Networks"
_ICLR.cc/2021/Conference — ICLR 2021 Poster_

### Official Review · AnonReviewer2 · 2020-10-27
**Nice empirical study on learning temporal logic formula with Transformer**

**Rating:** 6
**Confidence:** 4

**Review:**

This paper presents a dataset of linear-time temporal logic (LTL) formulas, which are generated by conjoining popular LTL specification patterns, and then investigates the learning capability of the Transformer over these LTL formulas.  The experimental evaluation shows that the Transformer predicts solution with high accuracy and sometimes even outperforms the classic solver used for providing training data.

- Quality: Though Transformers are fairly standard models, this paper shows an interesting application, i.e. solving LTL formulas, and the experimental evaluations are convincing and promising.
- Clarity: The writing is very clear and easy to follow.
- Originality: There is no technical innovation, but the presented dataset is new.
- Significance: This work demonstrates the surprising capability of Transformers on predicting solutions of LTL formulas, which suggests a competitive alternative approach for solving LTL formulas, and also motivates other logical reasoning applications of Transformer.

Questions:

Q1: Any particular reason for choosing "126"? Is it a representative size of LTL formulas in real-world verification tasks?

Q2: If the Transformer _truely_ learns the semantics of LTL formulas, the syntactic distribution of LTL formulas should not matter. Have you tried to train Transformer on LTLPattern126 and tested it on LTLRandom35?

Q3: Boolean satisfiability in the DNF form is trivial to solve. The propositional logic formulas seem quite close to DNF. Have you considered SAT solving in the CNF form? SAT competitions (e.g. https://baldur.iti.kit.edu/sat-competition-2017/index.php?cat=results) provide various kinds (e.g. industrial, random) of SAT benchmarks.


Minor typos:
page-2, "Where we focus " => "While we focus"

---

> ### Author Response · Authors · 2020-11-23
> **Reply to AnonReviewer2**
>
> We thank the reviewer for the detailed review.
>
> - ”Any particular reason for choosing "126"? Is it a representative size of LTL formulas in real-world verification tasks?”:
> We chose a size of 128 as power of two and left room for the extra start and end tokens.
>
> - Generalization between LTLRandom and LTLPattern:
> This is a great suggestion. We report on these experiments in the common reply.
>
> - “The propositional logic formulas seem quite close to DNF. Have you considered SAT solving in the CNF form?”:
> Note that it is not the goal of this work to compete with classical SAT solvers. The benchmark sets from the SAT competition are, unfortunately, too ambitious for now as their size is challenging for a plain Transformer architecture (many have >1M clauses). We think, however, that the reviewer made an interesting suggestion. We ran a first small experiment on propositional formulas in CNF of up to length 150 tokens, which, in fact, turns out to be way too easy for the Transformer (99.9% accuracy and 98.9% syntactic accuracy). We will set up an even larger experiment and are planning to incorporate the results into the paper.

---

### Official Review · AnonReviewer4 · 2020-10-28
**Impressive results using a straightforward application of modern learning techniques.**

**Rating:** 7
**Confidence:** 4

**Review:**


The paper explores the application of modern learning techniques (transformers and tree positional encodings) to the task of producing valid traces for a given LTL specification. A series of experiments are conducted to explore the generalization power of the proposed approach, both in terms of formula size and style of constraint. The work follows a recent trend of the application of deep learning techniques to logic-based settings, and the authors present (to the best of my knowledge) the first attempt of applying deep learning techniques to this particular task.

One weakness of the paper is that the details on the proposed model are extremely sparse -- the authors jump from background to data sets to evaluation, without ever pausing to describe the transformer+tree propositional encoding. We are left to infer from the parameter settings included in the appendix as to what was actually done.

On the other hand, the work seems to make a significant contribution empirically. In particular, the following claim from 5.1 is extremely exciting: "Our best model predicted correct solutions in 83% of the cases, taking on average 15s on a single CPU". The authors fall short of describing how they will turn an arguably unsound process for generating valid traces into one that is verifiably correct, but the initial evidence looks extremely promising.

Throughout the evaluation, there seems to be an over-emphasis on syntactic accuracy. There are countless ways to solve many LTL formulae, and very few for any given formula will be syntactically close. While it is an interesting phenomenon to explore in the context of other claims (such as the choice of structural representation in Fig 9), it is of very little interest to be able to map an LTL to one particular syntactic style of trace.

Another negative aspect is the claim to generalization. I'm on board with the generalization to larger instances (this seems like a wonderful result!), but the claim that using SAT encodings is indicative of generalization doesn't seem to be well supported. In particular, the very nature of the constraints has changed compared to the spot-generated benchmarks, and so there is little evidence from this evaluation that you haven't just learned to re-produce spot-like constraints. To show that level of generalization, I would expect to see a model trained using random LTL or benchmark LTL problems w/ spot, and testing done in a different (out-of-distribution) setting (like SAT or traces generated in an entirely different way than spot). Ultimately, I think that syntactic similarity is far less interesting than semantic similarity (i.e., correctness), and so I don't view this negative aspect of the paper to be substantial.

I am leaning towards accepting the paper. I would like to see a more complete exposition in terms of the DL model that was actually used, but the strength of the results, particularly in the larger problem sizes, is very promising.


1. You report on the "best model" results, but how do these compare to the average (mean or median)?

2. How would you embed what it is that you've done to create a sound / complete trace synthesis procedure?

3. How might you invert the process so that the LTL formula is synthesized that best captures a set of traces (or similarly describes the contrastive difference between two sets of traces)? E.g., the setting of "Bayesian Inference of Linear Temporal Logic Specifications for Contrastive Explanations" Kim et al.

A final point of recommendation (not affecting my decision on the paper). It's unclear how difficult the produced formulae are for producing a valid trace -- beyond that, it is unclear even how this might be measured. But one thing that is worth considering in the discussion part of the paper (if you have a clear sense of these aspects) is where th benchmark instances lay on the spectrum of easy-to-hard. If it is overly easy to produce a trace that satisfies the formula, then the results of the paper are less interesting (I doubt this is the case, given that standard solvers in the field are failing to find a solution). It would be interesting to know if the performance of the approach is a function of the problem "difficulty" (or space of possible solutions).

---

> ### Author Response · Authors · 2020-11-23
> **Reply to AnonReviewer4**
>
> We thank the reviewer for the detailed review.
>
> - ”[...] the details on the proposed model are extremely sparse -- the authors jump from background to data sets to evaluation, without ever pausing to describe the transformer+tree propositional encoding.”:
> We will add a more detailed description to the paper, if the reviewer finds this necessary. Note that we also published our code as supplementary material.
>
> - ”The authors fall short of describing how they will turn an arguably unsound process for generating valid traces into one that is verifiably correct, but the initial evidence looks extremely promising.” “How would you embed what it is that you've done to create a sound / complete trace synthesis procedure?”:
> Thanks for pointing this out! We will add a discussion on this to the paper. Checking if a trace satisfies an LTL specification can be checked very efficiently. A sound trace synthesis procedure could thus first query the Transformer for many predictions, and then check these predictions with efficient algorithms. If completeness is required, we can use classical methods as a fall back for cases where the Transformer did not produce correct answers.
>
> - ”Throughout the evaluation, there seems to be an over-emphasis on syntactic accuracy. [...] it is of very little interest to be able to map an LTL to one particular syntactic style of trace.”:
> We agree that semantic accuracy is the relevant measurement when we want to judge the performance of Transformers on this task. We will emphasize this in the intro to Section 5. In our paper we contrast semantic accuracy with syntactic accuracy as the training task is purely syntactic, and their comparison provides insight into what the Transformer learned.
>
> - ”[...] the claim that using SAT encodings is indicative of generalization doesn't seem to be well supported. [...] I would expect to see a model trained using random LTL or benchmark LTL problems w/ spot, and testing done in a different (out-of-distribution) setting (like SAT or traces generated in an entirely different way than spot).”:
> We do not claim that Transformers trained on LTL generalize to SAT. Our experiment on SAT was merely to support the claim that Transformers learn the semantics of (simple) logics rather than specifics of the generator. Since the SAT data used a very different generator (a CDCL solver vs automata constructions) this provides some evidence that the effect is independent from the specific generator we used for LTL.
>
> - “You report on the "best model" results, but how do these compare to the average (mean or median)?”:
> Even if we change the hyperparameters of the Transformers, the variance in the results is very small. In Appendix C we present results for different hyperparameters.
>
> - ”How might you invert the process so that the LTL formula is synthesized that best captures a set of traces (or similarly describes the contrastive difference between two sets of traces)? E.g., the setting of "Bayesian Inference of Linear Temporal Logic Specifications for Contrastive Explanations" Kim et al.”:
> This is a very interesting question, and, in fact, something we have on our agenda for future work.
>
> - ”A final point of recommendation (not affecting my decision on the paper). It's unclear how difficult the produced formulae are for producing a valid trace -- beyond that, it is unclear even how this might be measured.”:
> The LTLPattern formulas are difficult to solve. We constructed them in the same manner as used to evaluate LTL sat solvers, where this was the most challenging benchmark class.

---

### Official Review · AnonReviewer1 · 2020-10-28
**Promising approach, some questions remain**

**Rating:** 7
**Confidence:** 4

**Review:**

The paper presents multiple dataset generation and testing procedures for linear temporal logic and propositional logic satisfiability. They are then used to train Transformers with tree positional encoding. The approach amounts to imitation learning based on existing solvers for satisfiability for the considered logics. In contrast to previous work, the approach supports logical formulas of arbitrary shape. The experiments demonstrate successful generalization for multiple approaches to dataset generation.
Most importantly, the LTL model generalizes to examples that lead to timeout when using the imitated solver directly.

Strengths:
- Satisfiability is a prime use case for deep learning in rigorous formal methods, as solutions are hard to find but easy to check.
- Deep learning for LTL is interesting and I have not seen it done before.
- The models train and generalize successfully, they beat the existing approaches used for dataset generation on some instances.
- The authors attempted to exclude a strong dependency of generalization on their specific generation procedure by considering multiple different distributions for dataset generation.

Weaknesses:
- The training and test datasets (with the apparent exception of LTLUnsolved) are limited to easy instances, skewing the distribution. What happens if you use satisfiable examples where the generator times out after 1s as a test set?
- The distribution of LTLUnsolved is not discussed in detail. How exactly were those formulas obtained? Is it a similar procedure as the one used for LTLPattern126? If not, how does it differ? How do you make sure that the formulas are satisfiable?
- The paper does not show generalization between LTLRandom and LTLPattern datasets, even though this would be an obvious experiment to try if the goal is to demonstrate that the model learns LTL semantics in both cases instead of overfitting to the dataset generation procedure and the generator.


Detailed comments/questions:

Page 4:
Termination condition 4 requires solving a satisfiability instance after each new conjunct is added. How do you check this satisfiability and how does it relate to the 1s timeout?

Page 6:
"With this experiment we can already deduce that it seems easier for the Transformer to learn the underlying semantics of LTL than to learn the particularities of the generator."

Actually, I would also expect semantic accuracy to suffer quite a bit less than syntactic accuracy if the Transformer learns an imperfect copy of the generator, so it is not so clear whether the model learns semantics or just predicts imperfect sequences that coincidentally also work. (After all, your loss function instructs the model to imitate the generator.)
Maybe you could analyze the sensitivity of the generated traces to random noise. It is of course plausible that different aspects of multiple input/output-examples are successfully combined by the model in a way that leads to successful generalization even though it is distinct from what the generator does, but personally, I wouldn't claim that the model discovers a representation of the underlying semantics without some sort of demonstration of knowledge transfer to a vastly different LTL-related task.

Page 7:
The caption of Figure 3 claims that your model was trained on LTLUnsolved254 which seems impossible as that data set does not have output examples. In the text, you say that you trained on LTLPattern126, which makes more sense.

Page 8:
"Note that we allow the derived operators ⊕ and ↔ in these experiments, which succinctly represent complicated logical constructs." Those operators are not any more complicated than ∧ and ∨.


Minor:

Page 3:
- "significant improvements on a theorem proving"
- "(Selsam et al., 2019) presented". Use \citet{...} here.
- "(LTL) (Pnueli, 1977)". Maybe use \citep[LTL,][]{...}, here.

Page 7:
- "this experiments"

---

> ### Author Response · Authors · 2020-11-23
> **Reply to AnonReviewer1**
>
> We thank the reviewer for the detailed review.
>
> - LTLUnsolved: “How exactly were those formulas obtained? Is it a similar procedure as the one used for LTLPattern126? If not, how does it differ? How do you make sure that the formulas are satisfiable?”:
> LTLUnsolved254 is very similar to LTLPattern126 but consists of formulas of length up to 254 that Spot could not solve within 60 seconds. We will add a description of the LTLUnsolved254 dataset and the rationale behind its design to Section 3.
>
> - ”The training and test datasets (with the apparent exception of LTLUnsolved) are limited to easy instances, skewing the distribution.” ”What happens if you use satisfiable examples where the generator times out after 1s as a test set?”:
> We tested this with the LTLUnsolved254 dataset. Our models are trained on problems that took less than 1 second to solve with Spot (LTLPattern126) and it solves over 83% of the formulas for which Spot took >60 seconds (i.e. LTLUnsolved254). We chose this gap (<1s vs >60s) to keep data generation efficient for training but for testing we want to ensure that the models take less inference time than Spot would take to solve the formulas.
>
> - Generalization between LTLRandom and LTLPatterns:
> This question was also asked by reviewers 3 and 2, and we answered it in a general reply.
>
> - ”Termination condition 4 requires solving a satisfiability instance after each new conjunct is added. How do you check this satisfiability and how does it relate to the 1s timeout?”:
> Since just one of the 4 conditions must be met for termination, we do not have to distinguish between cases 3 and 4. We can stop Spot after 1 second irregardless of whether it would result in a trace or an unsat result. We clarified this in the paper.
>
> - ”[...] it is not so clear whether the model learns semantics or just predicts imperfect sequences that coincidentally also work.”:
> The construction of our datasets, especially LTLPattern126 and LTLUnsolved126, is designed to yield hard-to-solve formulas that have specific solutions that are hard to get right coincidentally. This is reflected in the timeout of the automaton construction of spot (and the timeout of other tools on such datasets [Li et al. 2013a]). The larger the state space of the automaton, the more specific a solution must be.
>
> - “Maybe you could analyze the sensitivity of the generated traces to random noise.[...] I wouldn't claim that the model discovers a representation of the underlying semantics without some sort of demonstration of knowledge transfer to a vastly different LTL-related task”:
> We are not sure how random noise could be introduced in this setting. The new experiments (requested by reviewers 3 and 2) to test the generalization from LTLPattern to LTLRandom and vice versa can be seen as a stress-test of the claim that Transformers learned the semantics of LTL. In these results we see that the gap between semantic accuracy and syntactic accuracy was even more dramatic.

---

### Official Review · AnonReviewer3 · 2020-10-28
**Teaching Temporal Logics to Neural Networks**

**Rating:** 5
**Confidence:** 3

**Review:**

This paper applies transformer models to learning linear-time temporal logic (LTL) and propositional logic. The results show that, when trained on large datasets of random formulas, transformer models can perform quite well on within-distribution held out test tasks, and when equipped with tree-positional encodings, they exhibit generalization to longer formulas.

Strengths:
-The paper is very clearly written
-Novel domain for neural approaches
-The results on the provided datasets are strong

Weaknesses:
I think the main weaknesses fall into three categories: novelty of the approach, analysis and comparison to baselines, and use of purely synthetic test data.

Novelty of the approach:
The approach is not novel, using a transformer model and a previously proposed tree-positional encoding scheme.
Novelty in the approach is not required, as long as the paper also contains experimental analysis which provides insight into either the technique or the problem studied, and the evaluation is thorough. However, I find that there are weaknesses in both these aspects.

Analysis and baselines:
The main paper reports results of a single model, and does not report results of any baselines, besides the length generalization results. If the paper claims that transformers specifically perform well on these problems, then comparing with alternate baselines (such as sequence or tree RNNs) is necessary. If the claim is instead that high capacity models in general can perform well on important LTL tasks, then I think that the use of only synthetic test data is a weakness (see below). In either case, I think further analysis of the conditions under which the model succeeds vs fails is warranted.

Synthetic data:
This work evaluates models only on randomly generated synthetic test data, which I view as a disadvantage. Although the paper demonstrates that training models on a large synthetic corpus provides good within-distribution test performance (as well as length generalization), it’s unclear how it would perform on natural data. I also don’t have a clear sense of how difficult the training and testing problems are relative to tasks relevant to people. Is it possible to collect a small non-synthetic test corpus of LTL formulas, to verify that the trained models can generalize to tasks relevant to people? The LTLPattern126 dataset is constructed from formulas from 55 LTL specification patterns identified from the literature. Can the models trained on these patterns generalize to other patterns from the literature? Similarly, could models trained on a subset of the 55 patterns generalize to the held-out patterns?

Because of the lack of baselines and analysis, and the use of purely synthetic test data, it’s difficult to evaluate the results of the paper. For this reason, coupled with the fact that the approach is not novel, I recommend a weak reject. However, I would be willing to raise my score if concerns about baselines, analysis, or synthetic data were addressed. In particular, evaluation on non-synthetic data would be very valuable.

Additional suggestions:
As stated above, I think more detailed experimental analysis would be very helpful, and could greatly strengthen the paper.
-How do transformers compare to other models, such as tree or sequence RNNs?
-What qualitatively (or quantitatively) distinguishes those formulas which a transformer can solve from those it can’t? Could insights here lead to proposed changes in architecture or data generation?
-What qualitatively (or quantitatively) distinguishes those formulas for which a transformer achieves syntactic accuracy from those for which it achieves semantic accuracy but not syntactic accuracy? Again, could insights here lead to proposed changes in architecture or data generation?
-Could a model trained on LTLRandom35 generalize to LTLPattern126 and visa versa?
-How do models with the sequence positional encoder perform on the LTLpattern126 and LTLUnsolved254 datasets?

Minor comments:
-I’d definitely recommend moving figure 9 to the main text, as it’s the only comparison between the model and a baseline.

---

> ### Author Response · Authors · 2020-11-23
> **Reply to AnonReviewer3**
>
> We thank the reviewer for the detailed feedback and suggestions.
>
> - “Analysis and baselines: The main paper reports results of a single model, and does not report results of any baselines [...]”:
> The main claim of our paper is that solving logical problems with neural networks works surprisingly well. Our baseline are the handwritten tools for LTL, such as Spot, and we show that we can solve some formulas faster than those tools. While it is an interesting question which architecture is best for logical reasoning tasks, we believe that this question has been covered in several other papers (e.g. Saxton et al. ICLR 2019), and Transformers clearly dominate in the settings we are aware of.
>
> - “This work evaluates models only on randomly generated synthetic test data”:
> To our knowledge there is no dataset of LTL formulas from applications. Previous works on LTL therefore introduced a distribution of LTL formulas based on the patterns found in applications. We used the same distribution to enable a fair comparison.
>
> - “I also don’t have a clear sense of how difficult the training and testing problems are relative to tasks relevant to people. Is it possible to collect a small non-synthetic test corpus?”:
> To put the results into perspective: It is sufficient to combine only three patterns for the specification of a simple arbiter (i.e., the request-response and mutual exclusion pattern). We combine up to 8 patterns in LTLPattern126 and up to 12 on the LTLUnsolved254. Classical tools start to struggle on formulas of this distribution after 9 patterns [Li et al. 2013a].
>
> - “Can the models trained on these patterns generalize to other patterns from the literature? Similarly, could models trained on a subset of the 55 patterns generalize to the held-out patterns?”:
> Following the reviewers suggestion, we did run various additional experiments. First, we created a dataset with larger random formulas, LTLRandom126, and trained a model on that data as well. Then we cross-tested our models, in particular those trained on LTLRandom126 and LTLPattern126, and also tested them on other (held-out) patterns from the literature. We will incorporate these results into the paper. A list of the patterns and references to the literature can be found at https://spot.lrde.epita.fr/genltl.html. LTLPattern126 was only constructed from the “dac” patterns to allow for comparison with the experiments from [Li et al. 2013a]. Results:
>
>
> Model trained on LTLRandom126, tested on other patterns:
> * “dac”  (i.e. the patterns we used to generate LTLPattern126)
>   - Semantic accuracy: 81.8%, 45 out of 55
>   - Syntactic accuracy: 49.1%
> * “eh”
>   - Semantic accuracy: 90.9%, 10 out of 11
>   - Syntactic accuracy: 81.8%
> * “hkrss”
>   - Semantic accuracy: 83.7%, 41 out of 49
>   - Syntactic accuracy: 71.4%
> * “p”
>   - Semantic accuracy: 90.0%, 18 out of 20
>   - Syntactic accuracy: 65.0%, 13 out of 20
>
> Model trained on LTLPattern126 tested on other patterns:
> * “eh”
>   -  Semantic accuracy: 36.4%, 4 out of 11
>   - Syntactic accuracy: 0.0%
> * “hkrss”
>   - Semantic accuracy: 49.0%, 24 out of 49
>   - Syntactic accuracy: 14.3%
> * “p”
>   - Semantic accuracy: 60.0%, 12 out of 20
>   - Syntactic accuracy: 10.0%
>
> - ”What [...] distinguishes those formulas which a transformer can solve from those it can’t?” and “What [...] distinguishes formulas for which a transformer achieves syntactic accuracy from those for which it achieves semantic accuracy but not syntactic accuracy?”:
> Thanks for the suggestion. We could not identify any further clear characteristics yet, other than that (which was no surprise) “overlapping untils” (e.g. (a U b) ∧ (a U ¬b)) are challenging to solve for the Transformer (so they are for classical solvers). We will add a short discussion on this into the paper.
>
> - ”Could a model trained on LTLRandom35 generalize to LTLPattern126 and visa versa?”:
> Since this was a common question, we answer it in a separate reply.
>
> - "How do models with the sequence positional encoder perform on the LTLpattern126 and LTLUnsolved254 datasets?":
> We are not exactly sure what experiment the reviewer is suggesting. Generalization from LTLRandom to LTLPattern126 or from LTLPattern126 to LTLUnsolved254 is very unlikely for models with the standard position encoding, given the results presented in Figure 9 in the appendix. We have not trained models with the standard position encoding on LTLPattern126. If the reviewer is interested, we can run this experiment. Training on LTLUnsolved254 is impossible as those formulas do not come with solutions.

---

### Author Response · Authors · 2020-11-23
**Common reply - OOD generalization**

A common question in the reviews was whether a model trained on random formulas could generalize to the pattern formulas and vice versa. Thanks for the interesting suggestion. We generated a new dataset LTLRandom126 to match the formula sizes and the vocabulary of LTLPattern126 and tested their OOD generalization with the following results:

Model trained on LTLRandom126, tested on LTLPattern126:
semantic accuracy: 24.7%, syntactic accuracy: 1.0%

Model trained on LTLPattern126, tested on LTLRandom126:
semantic accuracy: 38.6%, syntactic accuracy: 0.5%

Astonishingly, testing the models OOD increases the gap between syntactic and semantic correctness dramatically. This underlines the claim that the models learned something about the semantics rather than just the generator process. While the absolute accuracies are not impressive, we want to stress that the two distributions are very different.

We will add those experiments to the submission.

---

### Decision · Program_Chairs · 2021-01-07
**Final Decision**

**Decision:**

Accept (Poster)

**Comment:**

The paper shows that standard transformers can be trained to generate satisfying traces for Linear Temporal Logic (LTL) formulas. To establish this, the authors train a transformer on a set of formulas, each paired with a single satisfying trace generated using a classical automata-theoretic solver. It is shown that the resulting model can generate satisfying traces on held-out formulas and, in some cases, scale to formulas on which the classical solver fails.

The reviewers generally liked the paper. While the transformer model is standard, the use of deep learning to solve LTL satisfiability is novel. Given the centrality of LTL in Formal Methods, the paper is likely to inspire many follow-up efforts. There were a few concerns about the evaluation; however, I believe that the authors' comments address the most important of them. Given this, I am recommending acceptance. Please add the new experimental results (about out-of-distribution generalization) to the final version of the paper.